# Prevalence and Antimicrobial Resistance Profile of Diarrheagenic *Escherichia coli* from Fomites in Rural Households in South Africa

**DOI:** 10.3390/antibiotics12081345

**Published:** 2023-08-21

**Authors:** Phathutshedzo Rakhalaru, Lutendo Munzhedzi, Akebe Luther King Abia, Jean Pierre Kabue, Natasha Potgieter, Afsatou Ndama Traore

**Affiliations:** Department of Biochemistry and Microbiology, Faculty of Science, Engineering and Agriculture, University of Venda, Private Bag X5050, Thohoyandou 0950, South Africa; rakhalaru96@gmail.com (P.R.); lutendomunzhedzi1@gmail.com (L.M.); lutherkinga@yahoo.fr (A.L.K.A.); kabue.ngandu@univen.ac.za (J.P.K.); natasha.potgieter@univen.ac.za (N.P.)

**Keywords:** diarrheagenic, *Escherichia coli*, antibiotic resistance, households, kitchen cloths, toilets

## Abstract

Diarrheagenic *Escherichia coli* (DEC) pathotypes are the leading cause of mortality and morbidity in South Asia and sub-Saharan Africa. Daily interaction between people contributes to the spreading of *Escherichia coli* (*E. coli*), and fomites are a common source of community-acquired bacterial infections. The spread of bacterial infectious diseases from inanimate objects to the surrounding environment and humans is a serious problem for public health, safety, and development. This study aimed to determine the prevalence and antibiotic resistance of diarrheagenic *E. coli* found in toilets and kitchen cloths in the Vhembe district, South Africa. One hundred and five samples were cultured to isolate *E. coli:* thirty-five samples were kitchen cloths and seventy-five samples were toilet swabs. Biochemical tests, API20E, and the VITEK^®^-2 automated system were used to identify *E. coli*. Pathotypes of *E. coli* were characterised using Multiplex Polymerase Chain Reaction (mPCR). Nine amplified gene fragments were sequenced using partial sequencing. A total of eight antibiotics were used for the antibiotic susceptibility testing of *E. coli* isolates using the Kirby–Bauer disc diffusion method. Among the collected samples, 47% were positive for *E. coli*. DEC prevalence was high (81%), with ETEC (51%) harboring *lt* and *st* genes being the most dominant pathotype found on both kitchen cloths and toilet surfaces. Diarrheagenic *E. coli* pathotypes were more prevalent in the kitchen cloths (79.6%) compared with the toilet surfaces. Notably, hybrid pathotypes were detected in 44.2% of the isolates, showcasing the co-existence of multiple pathotypes within a single *E. coli* strain. The antibiotic resistance testing of *E. coli* isolates from kitchen cloths and toilets showed high resistance to ampicillin (100%) and amoxicillin (100%). Only *E. coli* isolates with hybrid pathotypes were found to be resistant to more than three antibiotics. This study emphasizes the significance of fomites as potential sources of bacterial contamination in rural settings. The results highlight the importance of implementing proactive measures to improve hygiene practices and antibiotic stewardship in these communities. These measures are essential for reducing the impact of DEC infections and antibiotic resistance, ultimately safeguarding public health.

## 1. Introduction

Diarrheal disease remains a significant public health issue particularly in rural areas where there is limited availability of clean water and adequate sanitation facilities [1,2]. The spread of pathogens through fomites is a serious concern to human health, safety, and development. Fomites act as reservoirs and potential vectors for pathogenic bacteria, including *E.coli*, leading to the spread of infections within households [3]. Pathogenic bacteria can survive on fomites for an extended period, and the duration of their survival is influenced by factors such as temperature, humidity, and the availability of other microorganisms [4,5]. Previous studies have shown the presence of *E. coli* on various fomites, including kitchen surfaces and cloths, toilet surfaces, door handles, and bathroom surfaces. These fomites serve as source of transmission, posing a potential health risk [6,7].

*E. coli* is a Gram-negative bacterium that typically inhabits the lower intestines of warm-blooded animals, and certain *E. coli*, O157: H7, leads to severe gastrointestinal infections in humans [8,9]. Studies by Seidman et al. [10] and Potgieter et al. [11] have reported the contamination of toilet seats in rural households with total coliforms and *E. coli*. Research findings have also revealed that kitchen cloths exhibit bacterial contamination, with *E. coli* emerging as the most frequently detected microorganism [9,12,13]. *E. coli* is generally used as an indicator of faecal pollution and indicates the presence of other pathogenic bacteria, such as *Salmonella* and *Shigella*, which have been associated with diarrhea [14]. Apart from its role as an indicator organism, *E. coli* can be classified as diarrheagenic (intestinal) or extraintestinal pathotypes [15]. Diarrheagenic *E. coli* pathotypes are categorized into six well characterized groups harboring specific genes: enteroinvasive *E. coli* (EIEC), enteropathogenic *E. coli* (EPEC), enterotoxigenic *E. coli* (ETEC), enterohemorrhagic *E. coli* (EHEC), enteroaggregative *E. coli* (EAEC), and diffusely adherent *E. coli* [16,17]. Certain *E. coli* pathotypes can acquire virulence genes from other *E. coli* strains, resulting in what is known as a hybrid pathotype [18,19,20]. Diarrhea caused by pathogenic *E. coli* is a leading cause of morbidity and mortality worldwide, especially in children younger than five years [21,22].

Infections caused by *E. coli* are usually treated using antibiotics such as penicillin, gentamycin, ampicillin, amoxicillin, chloramphenicol, rifampicin, and tetracycline [23]. However, some studies have documented that *E. coli* has become resistant to some antibiotics due to their widespread and inappropriate use [24,25]. Such misuse poses a serious health problem [26]; antimicrobial resistant *E. coli* strains have been reported as the main carriers of antibiotic resistance genes to ampicillin, penicillin, tetracycline, and rifampicin [27,28]. Hybrid DEC pathotypes have been reported to exhibit multidrug resistance to beta-lactam antibiotics [20,29].

In South Africa, inadequate access to water supply, sanitation services, and hygiene is considered the eleventh most significant risk factor leading to illnesses [30]. About 73% of toilets in rural households in the Vhembe District are pit holes with no water taps close to the toilets [31], suggesting that most people might not wash their hands immediately after using the toilets. Even in situation where water is accessible, most people wash their hands solely with water without using detergents [30].

Thus, poor sanitation and hygiene are still serious problems in rural households in the Vhembe district. Therefore, this study aimed to determine the prevalence and antibiotic resistance of diarrheagenic *E. coli* contamination in household fomites, highlighting the importance of implementing effective hygiene measures to mitigate transmission risks.

## 2. Methods and Materials

### 2.1. Study Area and Period

This study was conducted in Tshamutilikwa village (−22.892981245885206, 30.600267380011015) in the Vhembe district, South Africa (Figure 1), from May to August 2021. Tshamutilikwa is a place with a population of 814 people, according to the Census conducted in 2011 (https://census2011.adrianfrith.com/place/966110) (accessed on 3 August 2023). It covers an area of 1.06 square kilometers. With a population density of 766.49 people per square kilometer, Tshamutilikwa is a relatively densely populated area. The village consists of 203 households, resulting in an average of 191.15 households per square kilometer.

### 2.2. Ethics

Ethical clearance was obtained from the University of Venda [SEA/21/MBY/02/1608]. Sampling was done after receiving permission from the household owners by signing a consent form and questionnaire to answer and sign.

### 2.3. Sample Collection

Kitchen cloths, toilet seats, and toilet door handle swabs were collected door-to-door from the selected households and 70 toilet (35 seats and 35 door handles) samples were collected from participating households in Tshamutilikwa. A specific prepared questionnaire was administered to household owners to obtain information on various water, sanitation, and hygiene [WASH] factors, including the source of water, usage of kitchen cloths, condition of the kitchen cloth, toilet condition, type of toilet, handwashing means after using the toilet, the incidence of diarrhea in the household, and the practice of sharing the toilet with neighbors.

Samples were collected from toilet surfaces using the peptone water sterile swab-rinse method described by Hurst et al. [32]. In addition, a total of 35 old and used kitchen cloths were collected. Participants were requested to place the kitchen cloths in sterile zipping lock bags in exchange for new kitchen cloths. Before sample analysis, a brief description of the quality of kitchen cloths based on aspects such as dirty/clean or wet/dry was recorded. The toilets were categorized as clean or dirty based on their appearance. Clean toilets had no visible dirt, while dirty toilets had visible dirt, stains, and feces on toilet seats. The samples were immediately transported in ice to the microbiology laboratory and analyzed within four hours of sampling.

### 2.4. Bacterial Isolation and Identification

#### 2.4.1. Bacterial Isolation

In the laboratory, 5 cm by 5 cm (length × breadth) pieces were aseptically cut from each kitchen cloth sample and placed into a sterile flask containing 50 mL nutrient broth (Davies diagnostic (Pty) Ltd., Randburg, Gauteng, South Africa) for enrichment, vortexed for 5 min, and incubated at 37 °C overnight [33]. After incubation, 1 mL of the inoculated broth was transferred into a clean, sterile test tube containing 9 mL of sterile water. The diluted solution was mixed thoroughly by vortexing, and 0.5 mL of the diluted solution was then spread on sterile Eosin Methylene Blue (EMB) agar (Davies diagnostic (Pty) Ltd., Randburg, Gauteng, South Africa) plates using a sterile glass spreader and incubated for 24 h at 37 °C. Toilet seat and door handle swab samples were streaked directly on EMB agar and incubated for 24 h at 37 °C. After incubation, two distinct green metallic shiny colonies (characteristic of *E. coli*) were selected from each EMB plate and subcultured on a sterile nutrient agar plate to isolate pure colonies. All media used were prepared according to the manufacturer’s specifications.

#### 2.4.2. Bacterial Identification

The colonies obtained from sub-culturing on nutrient agar plates were analyzed using various biochemical tests such as the Kligler iron agar test [34], Urease test, Simmon citrate test [35], and the API20E (bioMérieux, Marcy I’Etoile, France). Presumptive *E. coli* isolates were further confirmed using the VITEK 2 automated system (bioMérieux, Marcy-l’Étoile, France) as described by the manufacturer. Briefly, a bacterial suspension was created by mixing *E. coli* colonies with 0.85% phosphate-buffered saline (PBS) (Thermo Fisher Scientific, Randburg, South Africa), resulting in a concentration of 1 × 10^8^ CFU/mL Mcflarland standard. Subsequently, 2 mL of these suspensions were automatically loaded into the VITEK 2 ID system, utilizing the GNB cards specifically designed for *E. coli* identification. The cards were analyzed through kinetic fluorescence measurement, and the results were reported within 3 h.

### 2.5. Molecular Identification of E. coli Isolates

#### 2.5.1. DNA Extraction

DNA extraction was performed as previously described by Omar et al. [36]. Briefly, 2 mL of nutrient broth with *E. coli* was aliquoted into sterile 2 mL Eppendorf tubes (Sigma-Aldrich, St Louis, MI, USA). The tubes were centrifuged at 13,000× *g* for 120 s to separate the cells from the supernatant. The DNA binding to celite was enhanced using lysis buffer mixed with 250 µL of 100% ethanol. Before washing, the celite-bound DNA was added to the spin columns. Qiagen elution buffer (Southern Cross Biotechnology^®^, Hilden, Germany) of 100 µL was used for DNA elution. Extracted DNA was then used as a template for PCR reactions.

#### 2.5.2. Genotypic Identification and Classification of *E. coli* Pathotypes

Genotypic identification and classification of selected isolates into the different *E. coli* pathotypes were performed using an 11-gene multiplex PCR, as previously reported [37,38]. The primers used in this study are in (Table 1). A total volume of 20 µL reaction mixture consisted of 10 µL, 2X Qiagen^®^ PCR multiplex mix (Qiagen^®^, Hilden, Germany), 1 µL 5× Q-solution, 2 µL of DNA template, 5 µL PCR grade water, and 2 µL of the primer mix containing 0.1 μM of *lt* and *mdh*, 0.5 μM of *stx1* and *st*, 0.3 μM of *eaeA* and *stx2*, and 0.2 μM of *astA, bfp eagg, ial,* and *gapdh* primers. Multiplex PCR amplification was performed in a Bio-Rad MyCycler^TM^ Thermal cycler (Bio-Rad, Hercules, CA, USA) under the following PCR conditions: an initial activation at 95 °C for 15 min, followed by denaturation at 94 °C for 45 s, and annealing was performed at 55 °C for 45 s. Extension was done at 68 °C for 2 min (35 cycles) [38]. PCR amplifications were separated using agarose gel electrophoresis, the bacterial DNAs were loaded into pre-cast wells in the gel, and a current was applied as described by Alfinete et al. [39].

### 2.6. Sequencing and Phylogenetic Analysis

Sequencing and phylogenetic analysis of *E. coli* was performed to compare the bacterial isolates obtained from the kitchen cloths and toilets within the same household and to investigate whether similar bacterial clones existed in different households, to identify any potential spread of identical clones within the community. DNA partial sequencing was performed on ABI 3500XL Genetic Analyzer POP7TM (Thermo Scientific, Waltham, MA, USA) using the same specific primers (Table 1). The reading of the DNA sequence was done and edited on FinchTV v1.4 (Geospiza Inc., Seattle, WA, USA). Nucleotide sequences of *E. coli* were compared with other reference strains on GenBank by blasting on the NCBI program (available at http://www.ncbi.nlm.nih.gov/) (accessed on 19 May 2022). For constructing the phylogenetic tree, MEGA X version 10.2.6 software was used to create phylogenetic trees by the neighbor-joining method and evaluated at 1000 bootstrap replicates for each gene [41,42].

### 2.7. Determination of Antibiotic Susceptibility

All the *E. coli* isolates were tested for sensitivity to different antibiotics using the Kirby–Bauer standard disc diffusion method [46,47]. For the disc diffusion assay, bacteria were grown for 24 h on Mueller–Hinton agar (Davies Diagnostics (pty) Ltd., Randburg, Gauteng, South Africa), harvested, and then suspended in 0.85% sterile PBS solution adjusted to a 0.5 McFarland turbidity standard, equivalent to 10^8^ CFU/mL. The standardized bacterial suspension was streaked onto Mueller–Hinton agar plates using a sterile cotton swab and exposed to commercially available antibiotic discs (Thermo Fisher Scientific, Waltham, MA, USA). The zones of inhibition were measured using a ruler after 24 h of incubation at 37 °C. The resistance patterns of the isolates to 8 different antibiotics (Table 2) were then interpreted as either Resistant (R), Intermediate resistant (I), or Sensitive (S), following the guidelines set by the Clinical Laboratory Standards Institute (CLSI, 2020) (https://clsi.org/meetings/susceptibility-testing-subcommittees/) (accessed on 3 August 2023). The antibiotics selected (Table 2) in this study are commonly used to treat diarrheal infections caused by diarrheagenic *E. coli* pathotypes.

## 3. Results

### 3.1. Study Characteristics

In all, 54.3% (19/35) of the kitchen cloths were dirty; among those, 54.6% (11/19) were dry and dirty, and 42.1% (8/19) were wet and dirty (Table 3). Of 35 toilets where swab samples were collected, 54.3% (19/35) were not clean, and two had feces on the seats.

### 3.2. Prevalence of E. coli on Kitchen Cloths and Toilets

Of the 105 samples collected, 46.7% (49/105) were positive for *E. coli.* All the kitchen cloths had bacterial contamination. A total of 71.4% (25/35) of the kitchen cloths (n = 35) were contaminated with *E. coli*. Out of the 70 samples collected from the toilets, 24 (34.3%) were contaminated with *E. coli* (Table 4).

### 3.3. Characterization of E. coli Pathotypes

The *mdh* gene was used as an internal control to ensure the PCR worked for each *E. coli* isolate. A total of 43/49 (90%) isolates were positive for the *E. coli* housekeeping gene (*mdh*). All the *E. coli* isolates with *mdh* genes also tested positive for the *gapdh* gene. The m-PCR test did not show any false positives or PCR inhibition as the external control gene (*gapdh*) was detected in all samples.

Multiplex PCR detected five DEC pathotypes (EAEC, EHEC, EPEC, ETEC, and EIEC). The prevalence of commensal *E. coli* (8/43; 18.6%) was lower than that of DEC (35/43; 81%). ETEC (22/43; 51%), harboring *lt* and *st* genes, was the most dominant DEC pathotype found in kitchen cloths and on toilet surfaces.

Different hybrid pathogenic strains of *E. coli* were found, 24 (55.8%) non-hybrid pathotypes and 19 (44.2%) hybrid pathotypes. There was a high prevalence of hybrids with two pathotypes, making the percentage 18.7% (Figure 2). Most of the hybrid *E. coli* strains exhibited the presence of the Asta gene, which is known to be carried by *E. coli* toxins.

Based on the sample type, more diarrheagenic *E. coli* pathotypes were found in kitchen cloths (39/49, 79.6%) followed by toilet seats (20/49; 40.8%) and toilet door handles (11/49; 22.4%), respectively (Figure 2). In addition, the gel image illustrating the results of the Multiplex PCR is provided in Appendix A.

### 3.4. Antibiotic Resistance Profile of E. coli

The *E. coli* isolates obtained from kitchen cloths exhibited varying levels of antibiotic resistance. *E. coli* isolates from the toilet surfaces and kitchen cloths displayed the highest resistance to ampicillin (24/24; 100%) and amoxicillin (24/24; 100%). (Table 5).

Some of the isolates showed resistance to more than two antibiotics; only 28.6% (14/49) of the isolates did not show resistance to three antibiotics. The multidrug resistance of *E. coli* isolates was found in 6.1% (3/49) of the isolates (Table 5). Only *E. coli* isolates with hybrid pathotypes were found to be resistant to more than three antibiotics used.

### 3.5. Sequence and Phylogenetic Analysis

#### Sequence Analysis and Phylogenetic Analysis

The study findings indicated that among the households sampled, only three households (numbers 7, 28, and 35) exhibited the consistent presence of the same pathotypes (ETEC, EHEC, and EAEC) across all three sample types: kitchen cloth, toilet seat, and toilet door handle surfaces. Specifically, household number 7 showed ETEC, 28 showed EHEC, and 35 showed EAEC detected in all sample types. Nine amplified DNA extracts were sent for partial sequencing (including three of *Stx1*, three of *lt*, and three of *Eagg*). Of the nine amplified *E. coli* isolates, only one *stx1* (1/3; 33.3%) and one *Eagg* (1/3; 33.3%) were successfully sequenced.

The two sequences obtained were blasted on GenBank for comparison with other reference *E. coli* strains. The similarities with the reference strain for the *Stx1* gene sequence ranged from 80 to 89.4%, and for the *Eagg* gene sequence ranged from 81 to 89.8% (Figure 3 and Figure 4).

The *Stx1* sequence (accession no. 0N193544) from the present study was closely related to a reference strain isolated in water from Hungary (accession no. DQ44966.1) and shared a common ancestor with an *E. coli* strain from human feces in Bangladesh (Figure 3).

The *Eagg* sequence (accession no. 0N241000) obtained from the toilet seat in this study shared a common ancestor with an *E. coli* strain (accession no. MZ330843.1) from handwash water in the Vhembe District, South Africa (Figure 4). There were limited reference strains of *Stx1* and *Eagg* genes in Africa.

## 4. Discussion

Poor sanitation and hygiene are still major problems in rural communities, especially in low- and middle-income countries (LMICs). They have been associated with increased diarrhea disease caused by enteric pathogens. Diarrhea is the leading cause of morbidity and mortality worldwide [42,48]. There are few reports on the prevalence and antibiotic resistance of diarrheagenic *E. coli* from household fomites in the Vhembe District. A large percentage difference was observed when comparing the bacterial rate of contamination on toilets (12/35; 34.3%) and kitchen cloths (25/35; 71.4%). This has been associated with the differences in the environmental conditions of the kitchen cloths (wet and dirty) and the toilets (dry and clean). The high frequency of *E. coli* on kitchen cloths can be attributed to the multipurpose use and the wet condition, which is suitable for the growth of bacteria [49,50]. The kitchen cloths examined in this study were highly contaminated with enterotoxigenic *E. coli*. Similar results were also reported by Chavatte et al. [12]. Furthermore, Speirs et al. [51] expressed concerns regarding the presence of enteric microorganisms in wet areas of the domestic kitchen, such as dishcloths, sink surfaces, and draining boards. These studies emphasize the potential health risks associated with dirty wet kitchen fomites that harbor bacterial contamination.

The high prevalence of DEC proves that kitchen cloths can be sources of food poisoning since ETEC, EHEC, and EPEC are pathogenic [33]. Studies have shown that the most frequently used fomites (toilets and kitchen cloths) are highly contaminated [52,53]. This study revealed a similar percentage of *E. coli* on both toilet seats and door handles. Similar findings have been previously reported [6,52,54,55]. Therefore, household toilets and kitchen cloths should be seen as important vehicles for transmitting diarrheagenic *E. coli* to humans.

Some of the presumptive *E. coli* isolates did not show the presence of the *mdh* gene, which could be due to the low DNA concentration or some PCR inhibitors and is in line with another study [40] in South Africa that reported that 15% of *E. coli* isolates were negative for the *mdh* gene. However, *E. coli* isolates that tested positive for *mdh* showed the presence of the *gapdh* gene, which was used as an external control. Using the *gapdh* gene as an external control helps ensure accurate PCR results with no false positives and no PCR inhibitors [40,56].

In this study, it was discovered that 44.2% of the *E. coli* isolate exhibited the combination of two or more genes from different pathotypes. Furthermore, hybrid pathotypes were more prevalent on kitchen cloths and toilet seats, respectively. Enterotoxigenic *E. coli* and EAEC were the most prevalent DEC in the diarrheal stool samples of young children living in the Vhembe district [22]. In addition, *E. coli* isolates with two or more virulence genes of DEC were found. Banda et al. [57] reported similar findings in the toilets and floor swabs from households in the Vhembe District, South Africa. The challenge concerning hybrid pathogens lies in their combination of virulence genes that leads to the development of severe diseases [58]. Several DEC strains with more virulence genes have been observed elsewhere in children’s diarrhea stool samples [48]. Previous studies reported an increase in the number of infections due to emerging DEC hybrid pathotypes [59,60,61]. Identifying a substantial proportion of diarrheagenic *E. coli* hybrid pathotypes on fomites highlights the need for effective hygiene measures in rural households. These findings highlight the potential for these fomites to serve as a reservoir for harmful bacteria, increasing household members’ risk of diarrheal illnesses.

A high prevalence of DEC resistance to commonly used antibiotics was found in the study area. Kitchen cloths and toilet surfaces in the rural areas of the Vhembe District were contaminated with DEC strains exhibiting high resistance to Beta-lactam antibiotics (ampicillin, amoxicillin, and penicillin). There is an increase in DEC resistance to amoxicillin and ampicillin in the current study as compared with the previous studies in Africa [60,61,62]. The inappropriate use of antibiotics has been identified as a contributing factor to antibiotic resistance in developing countries [10]. However, all the DEC isolates were susceptible to azithromycin and gentamycin. High *E. coli* susceptibility to gentamycin has been previously reported [63]. This study identified *E. coli* strains exhibiting resistance to multiple antibiotics. These findings agree with earlier reports on *E. coli* multidrug resistance in South Africa [64,65,66]. For example, Bolukaoto et al. [39], reported the multidrug resistance of DEC to two or more antibiotics (ampicillin, amoxicillin, cefotaxime, and others). This indicates a concerning situation where these commonly used antibiotics may not effectively treat infections caused by these resistant strains of DEC.

Sequences of *Stx1* and *Eagg* gene fragments identified in this study were related to reference strains associated with infections such as hemolytic uremic syndrome (HUS) and diarrhea in patients from Egypt, South Africa, China, Japan, USA, and the UK [66,67,68,69,70,71,72], suggesting that *Stx1* and *Eagg* strains found in this study may pose a threat to human health. The first outbreak of human *Stx1* disease in South Africa occurred in 1992, a decade after the first outbreak in the United States of America [73]. *Eagg* identified in this study from the toilet seat and *Eagg* previously isolated from handwash water in 2019 (accession no. MZ330843.1) in the Vhembe District share a similar ancestor (Figure 4). This shows inadequate hygiene and sanitation and possible routes of transmission from the toilet to humans. This reveals the continuous spread of diarrheagenic *E. coli* from 2019 to 2021 in the Vhembe District. Furthermore, Ojima et al. [65], and Sharma et al. [13] demonstrated that washing dishcloths with regular detergent or soaps was insufficient in destroying pathogenic bacteria and recommended soaking the dishcloths in sodium hypochlorite for 3 to 4 min, then washing them in hot water.

Enterotoxigenic *E. coli* and EAEC are significantly associated with hemolytic uremic syndrome (HUS), urinary tract infection, and diarrhea worldwide [66,67]. *E. coli* strains have been classified based on genetic and evolutionary relationships into four main phylogroups (A, B1, B2, and D). Some studies have reported that *E. coli* strains with *Stx1* and *Eagg* genes fall under phylogroup B2 and D, respectively [68,69]. The phylogenetic trees from this study revealed the relatedness of *Stx1* and *Eagg* from South Africa with others from different countries (Figure 3 and Figure 4). However, there are few reference *E. coli* strains with the same genes in Africa. Therefore, *E. coli* sequences from this study play a vital role in providing valuable epidemiological data specific to Africa. By analyzing these sequences, researchers can gain insights into the prevalence, distribution, and potential transmission patterns of these particular *E. coli* strains within Africa. This information is important in understanding and addressing the region’s public health implications associated with these strains.

## 5. Conclusions

There is a high prevalence of pathogenic and antimicrobial-resistant *E. coli* on kitchen cloths and toilet surfaces in the Vhembe District, South Africa. Kitchen cloths and toilets should be seen as important fomites for transmitting DEC. Furthermore, this study highlighted the inefficiency of regular detergents or soaps in eliminating pathogenic bacteria from kitchen cloths, emphasizing the need for proper hygiene practices such as soaking the cloths in sodium hypochlorite and washing them in hot water. The findings in this study indicate the urgency of implementing effective measures to combat antibiotic resistance and improve domestic hygiene practices in rural households to mitigate the spread of DEC.

## Figures and Tables

**Figure 1 antibiotics-12-01345-f001:**
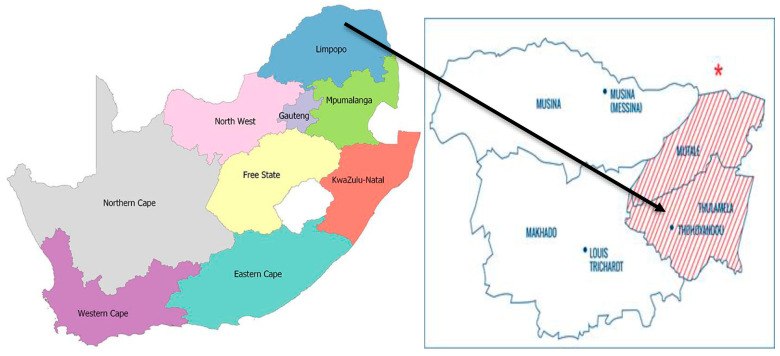
Map of South Africa (left) indicating sample collection site in the Vhembe District (Thulamela Municipality) where Tshamutilikwa village is located. The start indicates Vhembe District https://www.mappr.co/counties/south-africa/) (accessed on 3 August 2023).

**Figure 2 antibiotics-12-01345-f002:**
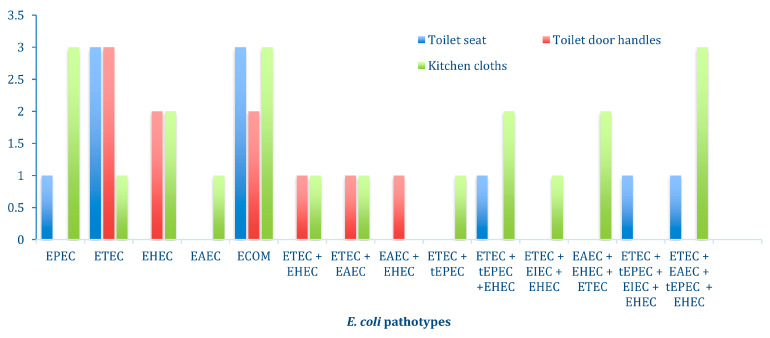
Prevalence of non-hybrid and hybrid *E. coli* pathotypes on kitchen cloths, toilet seats, and toilet door handles.

**Figure 3 antibiotics-12-01345-f003:**
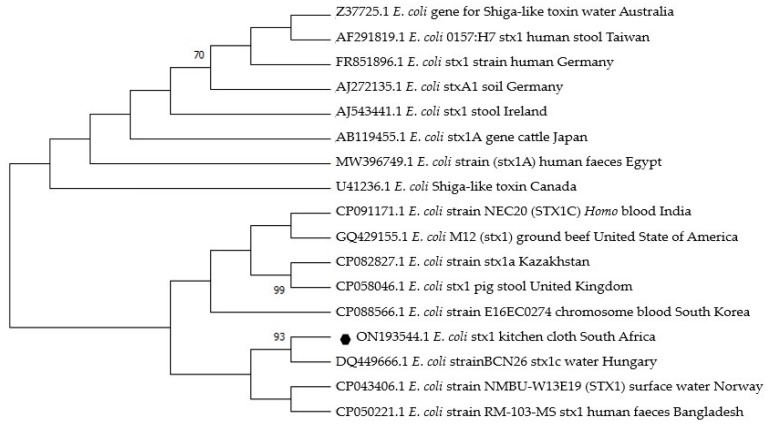
Phylogenetic tree based on 614 nucleotide sequences of the *E. coli* stx1 gene fragment constructed using the neighbor-joining method. The black dot indicates the sequence obtained from this study (May 2021) in the rural community of Vhembe District, South Africa. Fifteen reference *E. coli* strains with the same gene were selected randomly from GenBank. Bootstrap values greater than 70% for the branches were considered. The phylogenetic tree was constructed using Mega X version 10.2.6 software.

**Figure 4 antibiotics-12-01345-f004:**
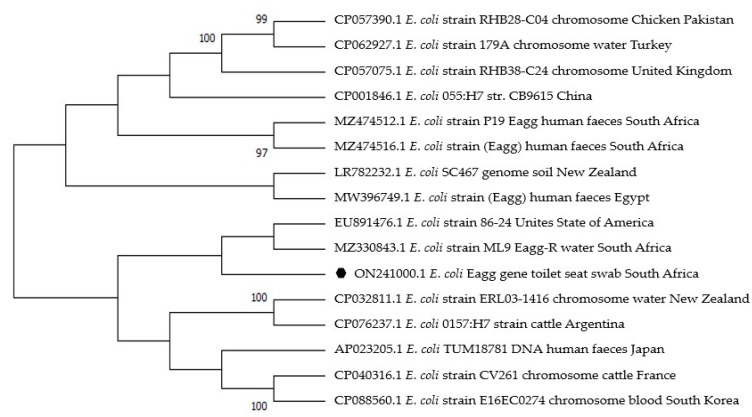
Phylogenetic tree based on 194 nucleotide sequence of *E. coli* eagg gene fragment constructed using the neighbor-joining method. The black dot indicates the sequence obtained from this study (May 2021) in the rural community of Vhembe District, South Africa. Fifteen *E. coli* reference strains with the same gene were selected randomly from GenBank. The phylogenetic tree was constructed using Mega X version 10.2.6 software.

**Table 1 antibiotics-12-01345-t001:** Primers used to identify diarrheagenic *E. coli* pathotype-associated genes.

Pathogen	Primers	Sequence (5′-3′)	Size (bp)	Conc. (µM)	Reference
*E. coli*	*mdh* (F)	GGT ATG GAT CGT TCC GAC CT	300	0.1	Omar et al. [40]
	*Mdh*(R)	GGC AGA ATG GTA ACA CCA GAG			
EIEC	*ial* (F)	GGT ATG ATG ATG AGT CCA	630	0.2	Pass et al. [41]
	*ial* (R)	GGA GGC CAA CAA TTA TTT CC			
EHEC/Atypical EPEC	*eaeA* (F)	GGT ATG ATG ATG ATG AGT CCA	917	0.3	Aranda et al. [42]
	*eaeA*(R)	GGA GGC CAA CAA TTA TTT CC			
Typical EPEC	*bfpA* (F)	AAT GGT GCT TGC GCT TGC TGC	410		
EAEC	*eagg* (F)	AGA CTC TGG CGA AAG ACT GTA TC	194	0.2	Pass et al. [41]
	*Eagg*(R)	ATG GCT GTC TGT AAT AGA TGA GAA C			
EHEC	*stx1* (F)	ACA CTG GAT GAT CTC AGT GG	614	0.5	Moses et al. [43]
	*stx1(R)*	CTG AAT CCC CCT CCA TTA TG			
	*stx2* (F)	CCA TGA CAA CGG ACA GCA GTT	779	0.3	
	*Stx2*(R)	CCT GTC AAC TGA GCA CTT TG			
ETEC	*lt* (F)	GGC GAC AGA TTA TAC CGT GC	330	0.1	Pass et al. [41]
	*lt* (R)	CGG TCT CTA TAT TCC CTG TT			
	*st* (F)	TTT CCC CTC TTT TAG TCA GTC AAC TG	160	0.5	
	*st* (R)	GGC AGG ATT ACA ACA AAG TTC ACA			
*E. coli* toxin	*astA (F)*	GCC ATC AAC ACA GTA TAT CC	106	0.3	Kimata et al. [44]
	*astA (R)*	GAG TGA CGG CTT TGT AGT C			
External Control	*gapdh* (F)	GAG TCA ACG GAT TTG GTC GT	238	0.1	Mbene et al. [45]
	*gapdh* (R)	TTG ATT TTG GAG GGA TCT CG			

**Table 2 antibiotics-12-01345-t002:** List of antibiotics used, disc potencies, and zone diameter interpretative standards for *E. coli* (CLSI, 2020).

Antibiotics	Disc Code	Disc Potency (µg)	Inhibition Zone (mm)		
			Resistant	Intermediate	Sensitive
Amoxicillin-Clavulanic	AMC	30	≤19	-	≥20
Azithromycin	AMZ	30	≤13	17–19	≥20
Chloramphenicol	CN	30	≤12	13–17	≥18
Gentamicin	GM	10	≤12	13–14	≥15
Ampicillin	AMP	10	≤13	14–16	≥17
Rifampicin	C	5	≤16	17–19	≥20
Tetracycline	TE	5	≤11	12–14	≥15
Penicillin	P	10	≤14	15–20	≥21

**Table 3 antibiotics-12-01345-t003:** Demographical features of the 35 households in Tshamutilikwa village with percentage.

Variables	Category	Total Study Population (%) (n = 35)
Number of people in a household	One	3 (8.6)
	Two	7 (20)
	More than two	25 (71.4)
Source of Water	Tap	33 (94.3)
	Well	0
	Surface	2 (5.7)
Handwashing means after using the toilet	Water only	17 (49)
	Water and soap	15 (43)
	Do not wash	3 (8.6)
Type of toilet	Ventilated improved latrine	27 (77.90)
	Pit latrines	1 (2.9)
	Flush toilets	7 (20)
Toilet condition	Clean	16 (45.7)
	Dirty	19 (54.3)
Sharing of the toilet with neighbors	Yes	2 (5.7)
	No	33 (94.3)
Animal ownership	Yes	29 (83)
	No	6 (17)
Animals are allowed to enter the house	Yes	1 (2.9)
	No	34 (97.1)
Diarrhea in the household	Yes	4 (11.4)
	No	31 (88.6)
Condition of kitchen cloth	Clean	16 (45.7)
	Dirty	19 (54.3)
Kitchen cloth use	Wiping up spills	2 (5.7)
	Drying hands	2 (5.7)
	Covering food	3 (8.6)
	Cleaning and drying up dishes	7 (20)
	Multi-use	21 (60)
Washing soap	Powdered soap	25 (71.4)
	Bar soap	6 (17)
	Jik bleach	4 (11.4)

**Table 4 antibiotics-12-01345-t004:** Prevalence of *E. coli* on kitchen cloths and toilet surfaces (seats and door handle).

Samples	*E. coli*	Percentage (%)
Kitchen cloths; n = 35	25	71.4
Toilet seats; n = 35	12	34.3
Toilet door handles; n = 35	12	34.3
Total; n = 105	49	46.7

**Table 5 antibiotics-12-01345-t005:** Antibiotic resistance percentage of *E. coli* isolated from kitchen cloths and toilets.

Resistance to Specific Antibiotic	Kitchen Cloths (51%) n = 25	Toilet Seats (24.5%) n = 12	Toilet Door Handles (24.5%) n = 12	Total (%) n = 49
Ampicillin	25 (100)	12 (100)	12 (100)	49 (100)
Tetracycline	4 (16)	1 (8.3)	0	5 (24.3)
Amoxicillin	25 (100)	12 (100)	12 (100)	49 (100)
Chloramphenicol	5 (20)	2 (16.6)	0	7 (36.6)
Rifampicin	3 (12)	0	0	3 (12)
Azithromycin	0	0	0	0
Gentamycin	0	0	0	0
Penicillin	19 (76)	7 (58.3)	9 (75)	35 (71.42)
Ampicillin and Amoxicillin	25 (100)	12 (100)	12 (100)	49 (100)
Ampicillin, Amoxicillin, and penicillin	19 (76)	7 (58.3)	9 (75)	35 (71.4)
Multidrug resistance Amoxicillin, Ampicillin, Penicillin, Chloramphenicol, and Tetracycline	2 (8)	1 (4)	0	3 (6.1)

## Data Availability

Raw sequencing reads are available in the National Center for Biotechnology Information (NCBI) GenBank database (accession numbers 0N241000 and 0N193544).

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
