# Peer review of "Prevalence and Antimicrobial Resistance Profile of Diarrheagenic Escherichia coli from Fomites in Rural Households in South Africa"

_antibiotics, 2023, doi:10.3390/antibiotics12081345_

Round 1
Reviewer 1 Report
This is an interesting manuscript, with several strengths. This article fits the scope of the journal and provides more data about the determination of the prevalence and antibiotic resistance of diarrheagenic E. coli from toilets and kitchen cloths. In order to straighten this article, points should be addressed accordingly before consideration in the journal:
Abstract
- Is too short, please expand. Results, conclusion and implications.
Material and methods
- Please give precise location codes.
- Can you provide more details on the VITEK 2 method?
- Please describe the disc diffusion method.
- Davies Diagnostic Pty, city, country) !!! Please indicate city and country.
Author Response
All comments were attended to

Reviewer 2 Report
The manuscript is overall well written. Hovewer, the reviewer have few comments:
1. It is now preferred for the authors to include at least some images of gels corraborating their results of the PCR reactions. They could be added as supplementary material.
2. What were the reference points used for determination of the resistance? What guidlines were followed? EUCAST? What were the growth inhibition zones?
English language requires minor spell check and grammar check, but the reviewer does not detect any major language issues.
Author Response
All comments were attended to

Reviewer 3 Report
comments are highlighted in attached file

Moderate editing of English language required
Author Response
All comments were attended to

Round 2
Reviewer 1 Report
Thank you for reviewing the manuscript.
Reviewer 2 Report
The authors have improved the manuscript and made necessary correction. The manuscript may be accepted in the current form
The English language has improved, it requires only minor editing and spell check